# The Possible Role of Apathy on Conflict Monitoring: Preliminary Findings of a Behavioral Study on Severe Acquired Brain Injury Patients Using Flanker Tasks

**DOI:** 10.3390/brainsci13020298

**Published:** 2023-02-10

**Authors:** Mariagrazia D’Ippolito, Giuseppe Spinelli, Marco Iosa, Salvatore Maria Aglioti, Rita Formisano

**Affiliations:** 1Department of Psychology, Sapienza University of Rome, 00185 Rome, Italy; 2Social and Cognitive Neuroscience Laboratory, IRCCS Fondazione Santa Lucia, 00179 Rome, Italy; 3Post-Coma Unit, IRCCS and Neuroreabilitation 2, Fondazione Santa Lucia, 00179 Rome, Italy; 4SmArt Lab, Laboratory for the Study of Mind and Action in Rehabilitation Technologies, IRCCS Fondazione Santa Lucia, 00179 Rome, Italy

**Keywords:** apathy, acquired brain injury, conflict monitoring, flanker tasks

## Abstract

The diagnosis of apathy, one of the most common behavioral changes after acquired brain injury (ABI), is important for improving clinical understanding and treatment of persons with ABI. The main aim of this study was to determine the possible role of apathy in conflict monitoring, by using choice reaction time tasks. **Methods**: We examined behavioral responses of conflict monitoring during three different flanker tasks in 10 severe ABI patients with or without diagnosis of apathy (3 M, mean age = 56.60; 3 M, mean age ± SD = 58.60, respectively), and 15 healthy controls (9 M, mean age = 54.46) Reaction times (RTs), accuracy, and global index of performance (GIP) were analyzed for each task. **Results**: Only apathetic ABI patients showed a significant difference from healthy subjects (*p*-value ≤ 0.001), while the performance of patients without apathy was not significantly different from that of healthy controls (*p*-value = 0.351). Healthy participants had higher accuracy in comparison to both ABI patients with (*p* < 0.001) and without (*p*-value = 0.038) apathy, whilst slower RTs were shown by ABI patients without apathy in comparison to both healthy subjects (*p*-value = 0.045) and apathetic ABI patients (*p*-value = 0.022). Only patients with apathy exhibited a significantly higher number of missing trials (*p*-value = 0.001). **Conclusions**: Our results may suggest a potential link between apathy following severe ABI and conflict monitoring processes, even though further investigations with larger sample size are needed.

## 1. Introduction

Apathetic manifestations are common across a wide variety of neurological and psychiatric conditions, such as traumatic brain injury (TBI) [1], disorders involving the basal ganglia [2,3], Alzheimer’s disease [4], and cerebrovascular accidents [5]. Arnould et al. [6], investigating the prevalence of apathy in TBI patients, revealed an overall point incidence of 47.3% of apathy across the studies they reviewed, while a study by Ciurli et al. [7] reported that TBI patients with a functional status recovery score indicating severe disability on the Glasgow Outcome Scale [8] had four times the risk of developing apathetic behaviors than TBI patients who have less severe scores.

Apathy is related to negative consequences both for patients and their caregivers [9], in terms of poor recovery [10], problems in daily functioning [11], financial and vocational loss, lack of post-injury social reintegration [12], and caregiver distress [13,14]. Yet, apathy is still a neglected neuropsychiatric syndrome in clinical practice, with no known standard treatment approaches, and it remains largely excluded from major psychiatric disease classification systems; on the other hand, apathetic manifestations in TBI populations often lead to more frequent and intensive consultations with healthcare centers and, therefore, represent a challenge to rehabilitation. There is no obvious relationship between the brain injury severity and the appearance of apathy [15].

Prigatano [16] described the psychosocial problems associated with lack of motivation, also termed “amotivation” or “adynamia”, in patients with ABI. Amotivation and adynamia are related to the negative symptoms of apathetic behavior and anhedonia [17], defined as a consistent and marked reduction of interest or pleasure in previously rewarding activities [18].

A potential source of confusion lies in the difficulty of clinically and conceptually differentiating apathy from depression, even though different studies have shown neuroanatomical and symptomatological differences between the two syndromes [19,20,21].

According to the World Health Organization’s international classification of diseases, depression is defined as a syndrome consisting in a persistent sadness (at least for two consecutive weeks) and in a marked diminished interest or pleasure and decreased energy, associated with at least one of the following symptoms: loss of confidence, excessive guilt, recurrent thoughts of death, poor concentration, sleep disorders, and change in appetite or weight. Furthermore, it can be also accompanied by alexithymia, characterized by both difficulties in identifying and describing one’s emotions and deficits in recognizing others’ emotional facial expressions.

Apathy is not a clinical criterion of depression, but it can be one of the clinical expressions of a depressive state [19,22,23]. The mechanisms by which depression induces apathy have not been totally clarified, even though it is very likely that apathy in depression results from an alteration of the emotional and affective processing via: (i) a marked sensitivity to emotionally negative situations inducing a negative bias interfering with attention resources and executive functions; or (ii) anhedonia (insensitivity to pleasure), which limits the will to perform actions.

In short, apathy is a symptom that can be observed in depression but may also occur without depression and, when both are present, they may be clinically and anatomically independent [19,21,24].

According to Marin [25], apathy is a lack of motivation, characterized by diminished goal-directed cognition (as manifested by decreased interests, a lack of plans and goals, and a lack of concern about one’s own health or functional status), diminished goal-directed behavior (as manifested by a lack of effort, initiative, and productivity), and reduced emotional concomitants of goal-directed behaviors (i.e., flat affect, emotional indifference, and restricted responses to important life events). Goal-directed behavior (GDB) is defined as a set of related processes (motivational, emotional, cognitive, and motor) by which an internal state is translated, through action, into the attainment of a goal [26,27], which can be immediate and physical, such as relieving thirst, or long-term and abstract, such as being successful in one’s job or pursuing happiness.

Other investigators emphasized that the absence of spontaneity observed among apathetic patients can be reverted under strong solicitation from the external environment, testifying to a contrast between a deep alteration of self-generated behaviors and a relative preservation of externally driven ones. Consequently, Levy and Dubois [28] defined apathy as the “quantitative reduction of self-generated voluntary and purposeful behaviors”, describing it as a pathology of voluntary action or GDB, and the underlying mechanisms responsible for apathy may be seen as dysfunctions occurring at the level of elaboration, execution, and control of GDB [27]. The authors divided apathetic syndrome into three subtypes (emotional, cognitive, and behavioral) but replaced the behavioral domain with the concept of auto-activation.

The “emotional–affective” subtype [28] refers to the inability to associate affective and emotional signals with ongoing and forthcoming behaviors. Any change in the linkage between emotion–affect and behavior may lead to apathy, either by reducing the willingness to perform actions (loss of will, loss of goals, emotional blunting) and maintain them to their completion or by diminishing one’s ability to evaluate the consequences of future actions [29]. It is due to orbital–medial prefrontal cortex (PFC) lesions [30,31], as manifested by a decreased impact of emotion and affect on ongoing or forthcoming behaviors.

“Cognitive apathy”, also named “cognitive inertia” [28], is the deficit in coordinating thoughts and actions with intentions to support social GDB, resulting in an impairment of elaborating a set of actions. It is related to executive functions’ impairments required to plan and carry out GDB, such as planning, working memory, and task switching.

A reduction of GDB can be secondary to lesions of the lateral PFC, which is represented by the dorsolateral (BA 9/46), ventrolateral (12, 44, 45, 47), and frontopolar (lateral 10) regions [32,33,34].

The “auto-activation” subtype, called “athymhormia” [28], refers to difficulties in activating thoughts or initiating the motor program necessary to complete the behavior. It consists in a loss of spontaneous activation that seems to affect both cognitive and emotional responses. Patients tend to remain quietly in the same place or position all day long, without speaking or taking any spontaneous initiative. Affect is usually flattened with anhedonia and emotional responses are blunted; any reactivity to emotional situations is poor and short-lived.

This syndrome has been reported after focal basal ganglia lesions [35,36,37,38,39], in most cases affecting, bilaterally, the internal portion of the pallidum [40,41,42,43,44,45]. It may also occur after frontal lesions affecting the frontal deep white matter (close to the medial PFC) [46]. Furthermore, Sultzer et al. [47] found that the auto-activation apathy symptoms are associated with low activity in bilateral insula.

There is some agreement within the literature that lack of interest, lack of initiative, and emotional blunting are all dimensions of apathy and that diminished GDB is at the core of the disorder [25,28]. Meanwhile, a number of studies suggested that depression, in particular major depressive disorder (MDD), is associated with deficits in cognitive control, specifically those involved in conflict monitoring [46,48,49,50]. Little is known about the relationship between apathy and conflict monitoring, especially in ABI patients.

Thus, in order to measure conflict monitoring and cognitive control in ABI patients with apathy vs. those without it, we employed one of most widely used interference tasks: the Eriksen flanker task [51]. It represents a recognized example of this response conflict, where subjects have to respond to a central target flanked by distractors, usually letters or arrows. When the target and flankers are the same (congruent condition), reaction time is shorter and performance is more accurate than when the target is different from the flanker (incongruent condition) [52].

Successful performance on this task, mainly on the incongruent condition, requires greater top–down cognitive control and a person’s ability to suppress inappropriate or prepotent responses [53], whereas unsuccessful performance has been reported in a number of clinical diseases such as schizophrenia, and substance use disorders and the abovementioned depression [54,55]. In our study, we used three different flanker tasks: the classic flanker task [51], where target and distractors were formed by letters, and another two modified versions.

Since apathy can be divided into three subtypes (i.e., “emotional–affective”, “cognitive”, and “auto-activation”; 28), our first modified task replaced the letters with emotional faces, while the second modified task substituted the letters with pictures of human hand postures, having the index finger pointing to right or left with a clenched fist. The modified emotional face flanker task may be linked to the emotional–affective subtype of apathy, while the flanker version with human hand postures could be associated with the auto-activation subtype, since we hypothesized that the hand image could have elicited the idea of action.

The first aim of the present study was to examine the relationship between apathy and conflict response in ABI patients, diagnosed with apathy, compared to those without apathy and healthy controls, by using the three different flanker tasks described above. Although recognizing the lack of studies in the literature concerning the conflict monitoring in ABI patients with diagnosis of apathy, we hypothesized that this clinical population would show deficit in conflict monitoring, exhibiting a worse performance with respect to ABI patients without apathy and healthy subjects, and showing a greater number of errors or missing responses.

The secondary aim of the study was to verify a possible correlation between the specific subtype of apathy (emotional–affective, cognitive, and auto-activation) and the type of flanker task (i.e., “cognitive apathy” vs. letter flanker task, “emotional–affective apathy” vs. emotional face flanker task, and “auto-activation apathy” vs. hand flanker task).

Given the lack in the literature of studies differentiating the subtypes of apathy, the aim of our study was to identify them, in order to develop targeted and effective rehabilitation programs aimed at decreasing the level of patients’ disability and improving their social participation.

## 2. Materials and Methods

### 2.1. Participants

Twelve severe ABI outpatients with diagnosis of apathy (ApABI) admitted at our post-acute neurorehabilitation hospital were screened. ApABI patients were included based on the following criteria: (i) age ≥ 18 years; (ii) diagnosis of severe ABI [56]; (iii) level of cognitive functioning (LCF) score ≥ 7 [57]; (iv) time interval from head trauma longer than 6 months. All patients underwent a cerebral computed tomography (CT) scan or magnetic resonance imaging (MRI) to confirm the traumatic or nontraumatic etiology of the ABI.

Participants were excluded from the sample in the case of (i) aphasia (score ≤ 29 in the token test) [58], (ii) any inability to undergo a formal psychometric assessment because of cognitive and/or severe sensory–motor deficits, (iii) previous/current history of psychoactive drugs and/or alcohol consumption/abuse, or (iv) previous history of psychiatric diseases and repeated ABI. Accordingly, 7 out of 12 participants were excluded (4 because of motor deficits, and 3 owing to hemi-spatial neglect and diplopia), resulting in a total sample of *5* ApABI patients (3 males, mean age ± SD = 56.60 ± 12.05 years). Five severe ABI patients without diagnosis of apathy matched for age and gender (3 males, mean age ± SD = 58.60 ± 11.60 years) were enrolled as a control group. Regarding the etiology of ABI, 6 patients had traumatic brain injury, whilst 4 vascular. They were selected from a database of volunteers at our neurorehabilitation institute, who previously gave their availability to take part in a research project.

All patients underwent a neuropsychological assessment administered by a trained neuropsychologist, consisting in the following tests and batteries: (i) Raven’s progressive matrices [59]; (ii) forward and backward digit span test [60]; (iii) Corsi block-tapping test [61]; (iv) prose memory test [62]; (v) frontal assessment battery (FAB) [63]; (vi) verbal fluency test [62]; (vii) trail making tests A and B (TMT_A, TMT_B) [64,65,66]

The diagnosis of apathy was assessed by first administering the clinician version of the Apathy Evaluation Scale (AES-C) [67] to each patient and then the NPI [68] to each patient’s caregiver (or family member). The AES-C is an 18-item instrument measuring apathy over the past 4 weeks and it is a reliable and valid measure of apathy following TBI, as it provides a multicomprehensive picture of both the cognitive and emotional–affective dimensions of apathy [1]. Each item (e.g., s/he gets things done during the day) is rated on a scale of 1 (Not at all characteristic) to 4 (Very characteristic) To control for the possible influence of depression, the Beck Depression Inventory (BDI) [69] was also used to assess levels of depressive symptoms in the patient samples. Due to the complexity of the apathy syndrome, it was difficult to distinguish the specific subtype of apathy exhibited by each patient; however, in our small sample, 2 patients were diagnosed as auto-activation apathy, 2 as emotional–affective, and one as cognitive, even though 3 of these patients showed symptoms amenable to all apathy subdomains.

All patients also filled in the Toronto Alexithymia Scale-20 (TAS-20) [70] to measure alexithymia, described as impairment in identifying personal emotions. Its 20-item revised version comprises three factors: (i) difficulty identifying feelings; (ii) difficulty describing feelings; and (iii) externally oriented thinking.

In addition, 15 healthy participants (9 males, mean age ± SD = 54.46 ± 9.57 years) were enrolled as a control group, having normal or corrected-to-normal vision, and without any previous or current neurological or psychiatric diseases, assessed by means of a psychological interview. BDI [69] and AES-C [67] were also administered to evaluate the possible presence of symptoms such as depression or apathy.

Each participant provided written informed consent prior to their participation. The study was approved by the local ethical committee and conducted in accordance with the standards of the 2013 Declaration of Helsinki.

### 2.2. Stimuli

The experiment consisted of three different tasks, all of them inspired by the flanker paradigm [51], but each defined by a specific set of stimuli (i.e., letters, human faces, and human hands).

The flanker task with letters (L-FT; Figure 1A) comprised white capital letters “H” and “S” as stimuli. Based on the nature of the flanker paradigm (i.e., a target stimulus flanked by two bilateral distractors), there were 4 possible conditions: (i) congruent condition: same target and same flanker (2 stimuli: HHHHH and SSSSS) and (ii) incongruent condition: one target and one different flanker (2 stimuli: SSHSS and HHSHH).

The face-flanker task (F-FT; Figure 1B), consisted in 2 different emotional face expressions (happy and sad) from the Karolinska Directed Emotional Faces database (KDEF; freely downloadable at http://www.emotionlab.se/resources/kdef (accessed on 1 September 2015) [71]. Eight types of face model were adopted: 4 male faces (2 happy and 2 sad) and 4 female faces (2 happy and 2 sad). Thus, the task consisted of 16 possible stimulus combinations representing 4 conditions: (i) congruent condition (2 stimuli: “happy target–happy flankers”, and “sad target–sad flankers”), and (ii) incongruent condition (2 stimuli: “happy target–sad flankers”, and “sad target–happy flankers”).

The hand-flanker task (H-FT; Figure 1C) consisted of pictures of a hand posture (right- or left-pointing index finger with a clenched fist) of 4 subjects (2 males and 2 females). As for the F-FT, this task included 4 possible combinations: (i) congruent condition (2 stimuli: index finger pointing to the right as both target and flankers, index finger pointing to the left as target and flankers), and (ii) incongruent condition (2 stimuli: index finger pointing to the right as target, index finger pointing to the left as flankers, index finger pointing to the left as target, index finger pointing to the right as flankers).

Each size class (307 × 105 pixels) stimulus (either a letter or a face or a hand) was presented on a black screen of a 15-inch computer monitor (1024 × 768 at 60 Hz), with a visual angle of 2° horizontally and 3.5° vertically. The visual angle between the center of the target and the center of each flanker was 0.5°. E-prime 2.0 (Psychology Software Tools) was used for stimulus presentation.

### 2.3. Procedure

Figure 2 depicts the timeline of the task. Participants sat on a comfortable chair in a quiet and well-lit room, at a distance of ~56 cm from the computer monitor. At the beginning of each trial, a fixation cross (“+”) was displayed for 600 ± 50 ms before the stimulus (lasting 600 ms) simultaneously showing both target and the flankers. To decrease expectancy effects, the fixation cross varied randomly between 600 ms, 650 ms, and 550 ms, with a mean fixation cross duration of 600 ms.

Participants were asked to respond to the target (in the L-FT: “H” or “S”; in the F-FT: “happy” or “sad” face expression; in the H-FT: “left” or “right” pointing direction) as quickly and accurately as possible, by pressing the corresponding key of a computer keyboard (“Q” or “P”) with their left or right index finger, respectively.

Each task (L-FT, F-FT, H-FT) consisted of a total of 480 trials (120 presentations of each stimulus). The occurrence of congruent and incongruent stimuli was presented in a randomized order and was counterbalanced across trials (50%). To control for any effect of response habituation, participants sequentially performed two sessions of each task (each lasting 240 trials, 60 presentations of each stimulus), in which the stimulus–response mapping was inverted. The order of sessions was counterbalanced within subjects, while the order of tasks was counterbalanced across subjects.

Before undergoing each of the three tasks, participants performed a practice section of 32 trials (8 trials for each stimulus).

### 2.4. Data Analysis

To evaluate the sample size, we planned to preliminarily acquire the data of 5 ABI patients with diagnosis of apathy, in order to compare their data with those of 15 healthy subjects. For our analyses, we fixed a threshold of 0.05 for the *p*-value to reject the null hypothesis and a power of tests at 80%. With these thresholds, we found statistically significant results with the group of 5 subjects without the need to increase the sample size with further acquisitions. Thus, we only planned to acquire the data of another 5 patients with ABI, but in this case without diagnosis of apathy.

Two measures of participants’ performance were considered for each task, i.e., reaction times (RTs) and accuracy (ACC). RTs were defined as the time interval between the onset of stimuli and the participant’s button pressing. To control for outliers, trials were excluded if the response time was more than 2.5 standard deviations (SDs) above or below the condition mean (a rule excluding less than 1% of responses). Moreover, trials were sorted according to the congruency of the stimuli presented (congruent vs. incongruent) and participants’ response (correct vs. incorrect response), for each block and each participant, separately. RTs and ACC (in %) were derived before computing a global index of performance (GIP), defined as the ratio between RTs (in ms) and ACC.

Data were firstly checked for normality (Shapiro–Wilk test) before computing parametric tests and post hoc tests for multiple comparisons, from the general linear models. Mann–Whitney U tests were used to compare, between the groups of ABI patients, the demographic and clinical features.

RTs, ACC, and GIP values were submitted to a mixed model analysis of variance (ANOVA) using Group as a between-subject factor (3 levels: healthy subjects, ApABI patients, ABI patients without apathy), and both Task (3 levels: L-FT, F-FT, H-FT) and Congruency (2 levels: congruent vs. incongruent) as within-subject factors.

Effect size was estimated by computing the partial eta squared (η^2^). Post hoc analyses were performed using Tukey’s correction on *p*-values. For all the analyses, the alpha level for significant results was set at 0.05.

Furthermore, the performances of both groups of patients (ApABI and ABI) were compared to those of the control group (healthy subjects), by means of a single-case study [72].

Crawford and Howell’s [72] method has been widely used to test for acquired deficits in single-case research [73,74,75,76,77] in order to detect how a patient’s score can depart from normality and to test the presence of deficits, regardless of the size of the control sample. In fact, Crawford and Howell’s test allows comparison of the patient’s performance with a modestly sized matched control sample. We chose, as a dependent variable, the difference in the GIP mean value between incongruent and congruent stimuli. These analyses were separately conducted for each task (L-FT, F-FT, H-FT). *p*-Values were corrected using Bonferroni–Holm’s procedure [78].

Finally, Spearman’s rank correlation coefficient was used to assess the possible relationship between the subtype of apathy (cognitive, emotional–affective, and auto-activation) and the performance of the apathetic patients in the 3 different tasks (L-FT, F-FT, H-FT). Scores on AES range from 18 to 72; the lower the AES score, the more it indicates the presence of apathy. Thus, clinical assessment of apathy showed significantly lower scores for ApABI patients with respect to both healthy subjects and patients without apathy.

## 3. Results

The mean interval in months from injury to date of assessment (chronicity) of ApABI patients was 60.4 ± 64.1, whilst that of ABI outpatients without diagnosis of apathy was 34.8 ± 23.8. Although this interval was lower than in ApABI patients, this difference was not statistically significant (*p*-value = 0.427, *t*-test) (Table 1). Age and education were compared among the three groups; no statistically significant differences were found for age (F = 0.317, *p*-value = 0.732), while significant differences were obtained for education (F = 4.436, *p*-value = 0.024) (Table 1).

Neuropsychological assessment and groups’ characteristics are reported in Table 2 and Table 3, respectively.

Clinical assessment of apathy was statistically significant among groups (F(2,22) = 124.06, *p*-value ≤ 0.001, η^2^ = 0.919), with significantly lower scores for ApABI patients in comparison to both healthy subjects and patients without apathy (post hoc analysis: *p*-value ≤ 0.001 for both), without any statistically significant differences between these last two groups (values are reported in Table 4). Comparisons of other clinical parameters between the two ABI patient groups showed statistically significant differences only for trail making test A (TMT_A). TMT_A was only significantly correlated to cognitive apathy (R = −0.773, *p* = 0.015).

No statistically significant difference was observed between the two groups of patients in the TAS-20 (Mann–Whitney U test: u = 11, z = −0.313, *p*-value = 0.754), nor was it found statistically correlated with neuropsychological scores (*p*-value > 0.05 for all clinical parameters).

### 3.1. Analysis on Performance (GIP)

Figure 3 shows the mean of GIP in all three groups of subjects, the three tasks, and the two conditions. ApABI patients showed higher values (i.e., worse performance) in the F-FT, and in the incongruent vs. congruent condition. A repeated measures ANOVA showed a statistically significant main effect for Group (F(2,22) = 9.196, *p*-value = 0.001, η^2^ = 0.455), Task (F(2,44) = 8.200, *p* = 0.001, η^2^ = 0.272), and Congruency (F(1,22) = 8.172, *p*-value = 0.009, η^2^ = 0.271), with lower values for congruent conditions. Post hoc analyses performed on Group revealed that only ApABI patients showed a significant difference from healthy subjects (*p* < 0.001), whereas the performance of patients without apathy was not significantly different from that of healthy controls (*p*-value = 0.351) who showed better performance. Regarding Task, subjects showed significantly better performances in the H-FT than in the F-FT (*p*-value = 0.009). The interaction between Group and Congruency only approached the significant threshold (F(2,44) = 2.904, *p*-value = 0.076, η^2^ = 0.209), whilst other interactions were even further from it. To deeply investigate these results, accuracy and RTs were also separately analyzed.

### 3.2. Analysis on ACC

Mean values of ACC are reported in Figure 4 which displays the higher accuracy of healthy subjects, the lowest accuracy of ApABI patients, and the lower accuracy for incongruent trials vs. the congruent ones. The analysis on the ACC highlighted significant main effects of Group (F (2,22) = 10.91, *p*-value = 0.001, η^2^ = 0.498) and Congruency (F(1,22) = 68.34, *p*-value < 0.001, η^2^ = 0.756), with higher accuracy for congruent vs. incongruent tasks. Post hoc analyses revealed that healthy subjects had higher accuracy in comparison to both ABI patients with (*p* < 0.001) and without (*p*-value = 0.038) apathy (Figure 4). No interaction effects resulted from the analysis.

### 3.3. Analysis on RTs

Slower RTs were observed in ABI patients without apathy, and again in the incongruent vs. congruent tasks. As for Task, faster RTs for H-FT in all the three groups were observed, as shown in Figure 5.

The analysis on RTs highlighted significant differences related to Group (F (2,22) = 4.71, *p*-value = 0.020, η^2^ = 0.300), and Task (F(2,44) = 11.67, *p*-value < 0.001, η^2^ = 0.347). Post hoc analyses revealed that the group effect was related to slower RTs of ABI patients without apathy in comparison to both healthy subjects (*p*-value = 0.045) and ApABI patients (*p*-value = 0.022). No significant differences were observed between healthy subjects and ApABI patients (*p*-value = 0.592) (Figure 5). As for Task, lower RTs (i.e., faster responses) in the H-FT with respect to both F-FT (*p*-value < 0.001) and L-FT (*p*-value < 0.001) were found.

Furthermore, post hoc analyses showed that the difference found in the interaction Congruency x Group (F (2,44) = 4.06, *p*-value = 0.032, η^2^ = 0.270) was related to significantly slower RT in the incongruent tasks of ABI patients without apathy in comparison to ApABI patients (*p*-value = 0.031). No further interaction effects were found.

Slower RTs in ABI patients without apathy were found vs. those with apathy (especially for incongruent trials); this unexpected result could be related to the fact that RTs were calculated without including missing trials, whose number was higher in ApABI patients than in the nonapathetic population (ApABI patients: L-FT, mean ± SD = 25.41% ± 69.32; F-FT, mean ± SD = 25.00% ± 70.88; H-FT, mean ± SD = 24.79% ± 74.06; ABI patients without apathy: L-FT, mean ± SD = 17.29% ±53.32; F-FT, mean ± SD = 22.70% ± 44.98; H-FT, mean ± SD = 11.45% ± 42.07; healthy subjects: L-FT, mean ± SD = 6.60% ± 32.46; F-FT, mean ± SD = 6.60% ± 27.57; H-FT, mean ± SD = 5.41% ± 28.52). To verify this hypothesis, a further analysis on missing trials was performed.

### 3.4. Analysis on Missing Trials

The number of missing trials was greatly higher in ApABI patients (mean: 120 ± 69) than in healthy subjects (30 ± 29), with the ABI patients positioned in the middle (82 ± 51). This difference was statistically significant among groups (F (2,22) = 9.320, *p*-value = 0.001, η^2^ = 0.459). Post hoc analyses showed a higher number of missing trials for ApABI patients with respect to healthy subjects (*p*-value = 0.001). Neither differences between patients with and without apathy (*p* = 0.359), nor between healthy subjects and ABI patients without apathy (*p* = 0.066) were statistically significant. Additionally, Congruency (F (1,22) = 51.334, *p*-value ≤ 0.001, η^2^ = 0.700) and Task (F (2,44) = 6.436, *p*-value = 0.004, η^2^ = 0.226) showed a statistically significant effect on missing trials, which were greater in the incongruent trials and face trials. Significant interaction effects were found for Group × Task (F(4,44) = 3.845, *p*-value = 0.009, η^2^ = 0.259) and Congruency × Task (F(2,44) = 3.289, *p*-value = 0.047, η^2^ = 0.130), while Group × Congruency (F(2,44) = 2.989, *p*-value = 0.071, η^2^ = 0.241) and Group × Task × Congruency (F(4,44) = 2.407 *p*-value = 0.064, η^2^ = 0.180) only approached the significant threshold (Figure 6).

### 3.5. Correlations with Apathy

A correlation between all the above analyzed parameters and the clinical assessment of apathy was tested. The total score of apathy was found significantly correlated with the ACC of L-FT incongruent trials (R = 0.414, *p*-value = 0.040), ACC of F-FT incongruent trials (R = 0.443, *p*-value = 0.026), and the number of F-FT missing incongruent trials (R = −0.449, *p*-value = 0.024).

Other significant correlations were found, by dividing the apathy score in the three main subtypes (emotional–affective, cognitive, and auto-activation), as shown in Table 5.

“Emotional–affective apathy” was transversally correlated with the subjects’ GIP. Indeed, it influenced many other parameters, such as: (i) number of missing trials in the incongruent condition of L-FT (R = −0.472, *p*-value = 0.017), in the congruent (R = −0.398, *p*-value = 0.049) and incongruent conditions of F-FT (R = −0.495, *p*-value = 0.012), in the congruent (R = 0.462, *p*-value = 0.020) and incongruent conditions of H-FT (R = −0.474, *p* = 0.017); (ii) ACC of F-FT incongruent trials (R = 0.482, *p*-value = 0.015), and of H-FT congruent (R = 0.494, *p*-value = 0.012) and H-FT incongruent trials (R = 0.462, *p*-value = 0.020); (iii) RTs of F-FT incongruent trials (R = 0.441, *p*-value = 0.027).

GIP was found significantly correlated with the “auto-activation apathy” in the congruent trials of H-FT (R = −0.403, *p* = 0.046). “Auto-activation apathy” was also related to the number of missing trials in both congruent (R = −0.570, *p*-value = 0.003) and incongruent conditions of H-FT (R = −0.582, *p*-value = 0.002).

“Cognitive apathy” was found significantly correlated only with ACC in F-FT incongruent trials (R = 0.418, *p*-value = 0.038). No other correlations were found between cognitive apathy and GIP (Table 4).

### 3.6. Single-Case Study

As for the L-FT, only one ABI patient without apathy reported a GIP incongruent–GIP congruent difference that exceeded the upper limit of the 95% confidence interval of the control group (Subject 23, Crawford–Howell *t*-test = 9.218, *p*-value ≤ 0.001 Bonferroni–Holm corrected), as well as two ApABI patients (Subject 18, Crawford–Howell *t*-test = 5.627, *p*-value ≤ 0.001 Bonferroni–Holm corrected; Subject 20, Crawford–Howell *t*-test = 19.897, *p*-value ≤ 0.001 Bonferroni–Holm corrected), whereas another two patients with apathy reported an inverse effect, obtaining a value below the 95% confidence interval of the control group (Subject 16, Crawford–Howell *t*-test = 7.640, *p*-value ≤ 0.001 Bonferroni–Holm corrected; Subject 19, Crawford–Howell *t*-test = 11.781, *p*-value ≤ 0.001 Bonferroni–Holm corrected) (Table 6; Figure 7). These findings may suggest that ABI patients with and without apathy that exceeded the 95% confidence interval of the control group were those that obtained higher GIP scores (i.e., worse performance) in the incongruent trials of L-FT and, thus, they could be more influenced by the flanker effect. Conversely, the two patients with apathy with mean values below the interval confidence of the control group were those with higher GIP scores in the congruent trials of L-FT and, consequently, they were less influenced by the conflict produced by flanker stimuli.

Regarding F-FT, three ABI patients without diagnosis of apathy fell below the confidence interval of the control group (Subject 21, Crawford–Howell *t*-test = 11.99, *p*-value ≤ 0.001 Bonferroni–Holm corrected; Subject 22, Crawford–Howell *t*-test = 4.984, *p*-value ≤ 0.01 Bonferroni–Holm corrected; Subject *23*, Crawford–Howell *t*-test = 7.841, *p*-value ≤ 0.001 Bonferroni–Holm corrected), since their GIP value, due to the difference between GIP values of incongruent and congruent stimuli, suggests a better performance in the incongruent trials (i.e., lower scores) in comparison to the congruent ones. Conversely, in the ApABI sample, three patients were above the 95% confidence interval of the control group (Subject 16, Crawford–Howell *t*-test = 4.679, *p*-value ≤ 0.01 Bonferroni–Holm corrected; Subject 19, Crawford–Howell *t*-test = 12.147, *p*-value =< 0.001 Bonferroni–Holm corrected; Subject *20*, Crawford–Howell *t*-test = 14.444, *p*-value ≤ 0.001 Bonferroni–Holm corrected) (Table 7; Figure 8). 

Finally, in the H-FT, only one ABI patient without apathy showed a GIP incongruent–GIP congruent difference above the confidence interval of the healthy control group (Subject 21, Crawford–Howell *t*-test = 4.167, *p*-value ≤ 0.01 Bonferroni–Holm corrected), as well as two ApABI patients (Subject 16, Crawford–Howell *t*-test = 8.954, *p*-value ≤ 0.001 Bonferroni–Holm corrected and Subject 19, Crawford–Howell *t*-test = 10.474, *p*-value ≤ 0.001 Bonferroni–Holm corrected). One ApABI patient also reported an inverse effect, obtaining a value below the 95% confidence interval of the control group (Subject 18, Crawford–Howell *t*-test = 3.403, *p*-value ≤ 0.05 Bonferroni–Holm corrected), and suggesting the presence of higher GIP scores (i.e., worse performance) in the congruent trials of H-FT and, consequently, a reduced flanker interference (Table 8, Figure 9).

## 4. Discussion

The aims of this study were to investigate the relationship between apathy and conflict monitoring in ApABI patients, compared to those without apathy and healthy controls, and to verify a possible correlation between the specific subdomain of apathy and the type of flanker task proposed

Very little is known about the potential influence of apathy on cognitive control (more specifically on conflict monitoring) and their relationship is still controversial. According to Andersson and Bergedalen [79], there was a significant association between more severe apathy and executive dysfunction, and different studies have found an association between apathy symptoms and poor performance on standard executive function tests [24,80]. For these reasons, apathy is frequently conceptualized as a “dysexecutive syndrome” [81,82]. However, some studies have revealed inconsistent results on the relationship between apathy and executive deficits, suggesting that executive function deficits are not crucial for the presence of apathy symptoms [82,83].

One of the typical “interference” paradigms used to measure executive control and examine conflict monitoring is the flanker task [51]; different versions of it exist, even though they share the same structure where participants have to recognize the centrally presented stimulus flanked by two bilateral distractors, which can appear either identical to the target (congruent condition) or different from it (incongruent condition). Usually, RTs are slower and ACC is lower in the incongruent condition owing to the interference related to the confusing flankers [84].

The main result of our study is that the performance (GIP) of ApABI patients was worse than that of healthy subjects, mainly in the incongruent trials, while that of ABI patients without apathy was not. This may support the notion that apathy, like anxiety and depression, can directly impact cognitive performance, in particular that related to conflict monitoring, masking the subject’s true ability [85]. On the other hand, as single-case analysis pointed out, some ABI patients with and without apathy showed worse performance in the incongruent trials of L-FT, suggesting that both clinical conditions (i.e., with or without apathy) can be influenced by the flanker effect. Indeed, the small sample size may influence the research finding, suggesting that the results be interpreted with caution.

The average reaction time of ApABI patients was not slower than that of patients without apathy, as expected; this may be due to the fact that missing trials were not included in the computation of RTs, as instead performed in the ACC calculation. In fact, only ApABI patients showed a significantly higher number of missing trials in comparison to healthy subjects, whereas patients without apathy did not; furthermore, the number of correct responses provided by ApABI patients was significantly lower than both healthy subjects and ABI patients without apathy. The significant interaction Group per Task also revealed that ApABI patients exhibited a higher number of missing responses in the F-FT, suggesting that this clinical population presented more difficulties in recognizing emotional face expressions than nonapathetic ABI patients and healthy participants. In fact, as highlighted by Njmboro and Deb [86], emotion recognition is usually impaired in patients with apathy.

These first results could support the main hypothesis of the study that ApABI patients may have had more difficulties in identifying the target stimuli, especially when target and distractors were facial emotion expressions. The association between the significantly higher number of missing trials and the faster RTs of ApABI patients in comparison with those without apathy seems to suggest a waiver strategy of apathetic persons with ABI. However, the 600 ms temporal window given to participants did not allow us to correctly evaluate if they responded after the target disappeared or failed to react to the stimulus, making errors of omission.

The secondary hypothesis of this study is that performance could be related to different flanker tasks in relation to the most severely affected domain of apathy; cognitive apathy may mainly affect the L-FT, emotional–affective apathy could influence the F-FT, and auto-activation apathy the H-FT. However, this hypothesis was only partially supported by data. Cognitive apathy showed a poor influence on the subjects’ responses. Conversely, emotional–affective apathy revealed a transversal effect on all types of tasks; results show that the more severe the emotional affective apathy the greater the number of missing responses.

A specific effect was found only for “auto-activation” apathy that was significantly correlated with GIP in congruent trials of H-FT, and with the number of H-FT congruent and incongruent missing trials. This result partially supports the hypothesis that the subdomain of apathy could be related to the specific type of flanker task since, in this case, auto-activation apathy was associated with the H-FT, with worse responses for patients showing this specific apathy subtype. However, these findings suggest that it is possible that apathy symptoms may impair performance on any task, as a result of a general lack of motivation shown by these patients.

However, regarding the H-FT, it is important to underline that participants sequentially performed two sessions of each task where the stimulus–response mapping was inverted. Consequently, subjects were asked to press key “P” when the target showed the right-pointing direction during a session, and “Q” for the left-pointing direction during another one. This may have led to a spatial compatibility effect, by contributing to a better performance in terms of GIP and RTs by the three groups, in this specific task.

The two groups of patients were well matched for all the analyzed demographical and clinical parameters. Only TMT_A was significantly different among the two groups, but it was found related to cognitive apathy, a parameter which poorly affects subjects’ performances. Thus, it can be deduced that attention, one of the cognitive processes explored by TMT_A, which, in turn, was found related to cognitive apathy, could have a poor influence on subjects’ responses.

Finally, given the higher number of missing responses showed by ApABI patients in all the three tasks, mainly in incongruent trials, the results may suggest a potential link between apathy following severe ABI and conflict monitoring processes, even though further investigations are needed.

## 5. Limitations

This study was conducted under some constraints and the major limitation is the small sample size, given the difficulty of enrolling ABI patients both with diagnosis of apathy and who did not show sensory–motor deficits (e.g., hemi-spatial neglect or visual disorders), that would make it impossible to execute the tasks. The small sample size limits the generalizability of findings to larger patient populations and the ability to determine the substantial role of apathy in conflict monitoring. For this reason, the results can only be interpreted with caution. However, despite the small sample size, many significant differences have been found between ApABI patients and the other two groups, suggesting that a wider sample will confirm these results, providing more solid ones.

Another limitation is related to the spatial properties (left–right pointing) of the H-FT which could represent a limitation of the task, since they may have caused some degree of spatial orienting of attention, not present in the other conditions, with a potential confounding effect on performance of patients.

Finally, the lack of information regarding the type and location of brain injury may also represent a constraint of the study, although the etiopatogenesis of apathy was beyond the scope of the study. It is interesting to note that ApABI patients seemed to exhibit greater visuo-spatial attentional/processing speed deficits (i.e., TMT-A completion time) than ABI patients without apathy, suggesting a fronto-parietal (-subcortical) impairment that may contribute to apathy and, potentially, to different apathy subtypes differentially.

## 6. Conclusions

Apathetic manifestations are commonly reported in the ABI population and have been considered to be one of the greatest barriers to reintegration into the community, affecting motivation to engage in rehabilitation, and have been also associated with a wide range of negative consequences for the patients and their caregivers [87]. This pilot study represents a first step in understanding the apathetic symptoms which impact significantly on patient quality of life, suggesting that a more routine assessment of apathy is required, mainly to discriminate it from depression.

The strength of this study, which we believe allows distinguishing our study from the previous ones, is the implementation of the H-FT. It has been based on the hypothesis that the “communicative gesture” of pointing could have activated the thoughts necessary to spontaneously initiate the motor program, usually compromised in patients with auto-activation deficit.

Further studies with a larger sample size should be carried out in future, in order to test the generalizability of the present results. It could be important to supply more evidence for the cortical processing of conflict control, by means of event-related potential recordings. More specifically, it could be interesting to address the neural correlates of cognitive control on affective conflicts, to reveal the possible interplay between conflict control and facial expression perception in ABI patients with diagnosis of apathy. For instance, some studies on emotional F-FT [88] revealed that when the target is friendly and distractors are angry, the flankers attract more attention away from the target, producing conflict, while when the target is negative (angry face) and flankers are positive (happy faces) attention is attracted to the target and away from the flankers, reducing possible conflicts. This result indicates that the negative target stimulus narrows the focus of attention, whereas positive stimuli may broaden it.

Further studies are also necessary to better identify the underlying mechanisms of apathy in order to develop targeted and effective rehabilitation programs, decrease the level of disability, and improve social participation.

## Figures and Tables

**Figure 1 brainsci-13-00298-f001:**
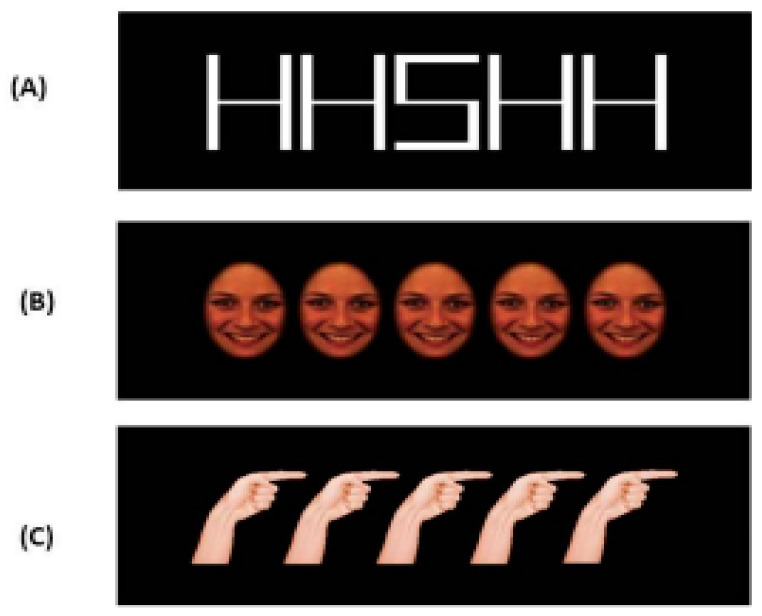
Stimulus examples. (**A**) The stimulus example in the incongruent condition in L-FT; (**B**) the congruent condition in F-FT (happy target–happy flankers); (**C**) the congruent condition in H-FT.

**Figure 2 brainsci-13-00298-f002:**
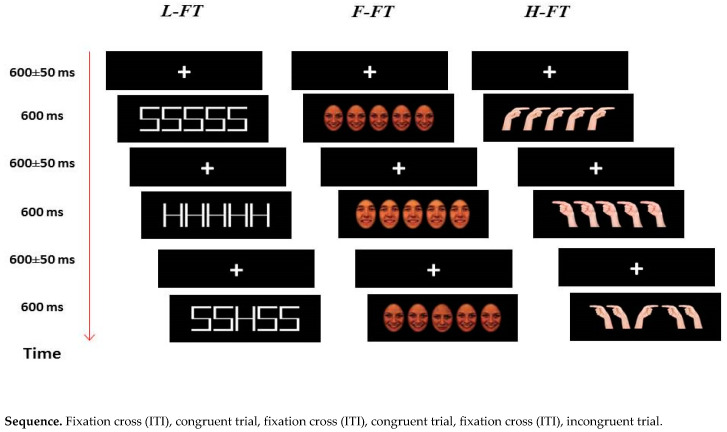
Timeline of the task.

**Figure 3 brainsci-13-00298-f003:**
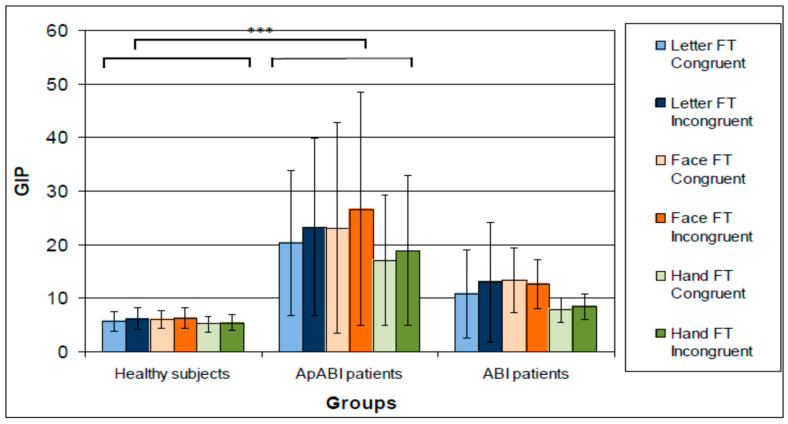
Mean of the GIP (higher values correspond to worse performance) in healthy subjects, ApABI and ABI patients for letter (blue), face (orange), and hand (green) FT, in congruent (light color) and incongruent (dark color) trials. *** stand for *p* ≤ 0.001.

**Figure 4 brainsci-13-00298-f004:**
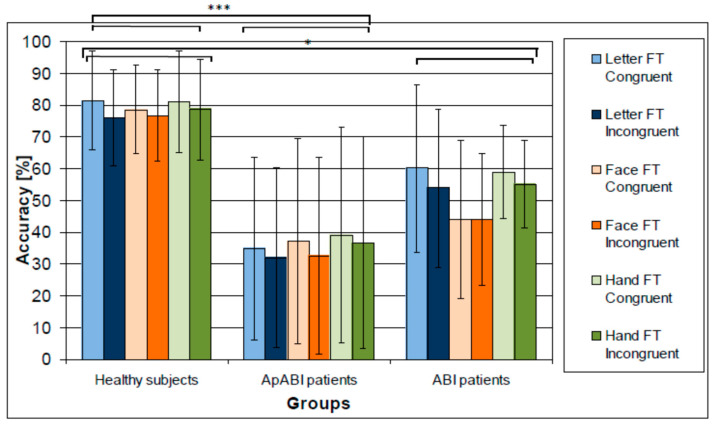
Mean of the ACC (higher values correspond to better accuracy) for healthy subjects, ApABI and ABI patients in the letter (blue), face (orange), and hand (green) FT, and in congruent (light color) and incongruent (dark color) trials. * stands for *p* ≤ 0.05; *** stand for *p* ≤ 0.001.

**Figure 5 brainsci-13-00298-f005:**
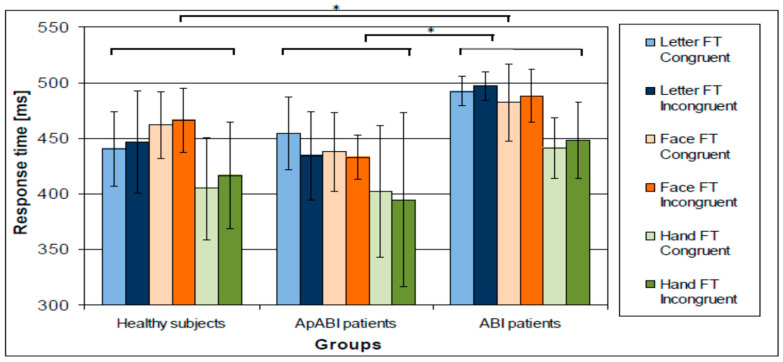
RT mean (higher values correspond to slower responses) for healthy subjects, ApABI and ABI patients in the letter (blue), face (orange), and hand (green) FT, in the congruent (light color) and incongruent (dark color) trials. * stands for *p* ≤ 0.05.

**Figure 6 brainsci-13-00298-f006:**
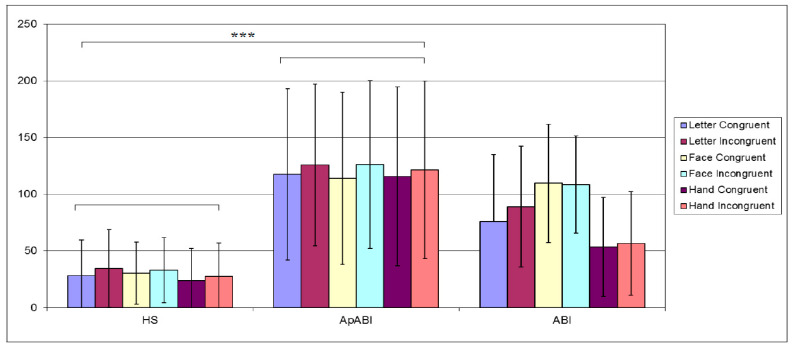
Number of missing trials in healthy subjects, ApABI and ABI patients for the L-FT, F-FT, H-FT, and both congruent and incongruent trials. *** stand for *p* ≤ 0.001.

**Figure 7 brainsci-13-00298-f007:**
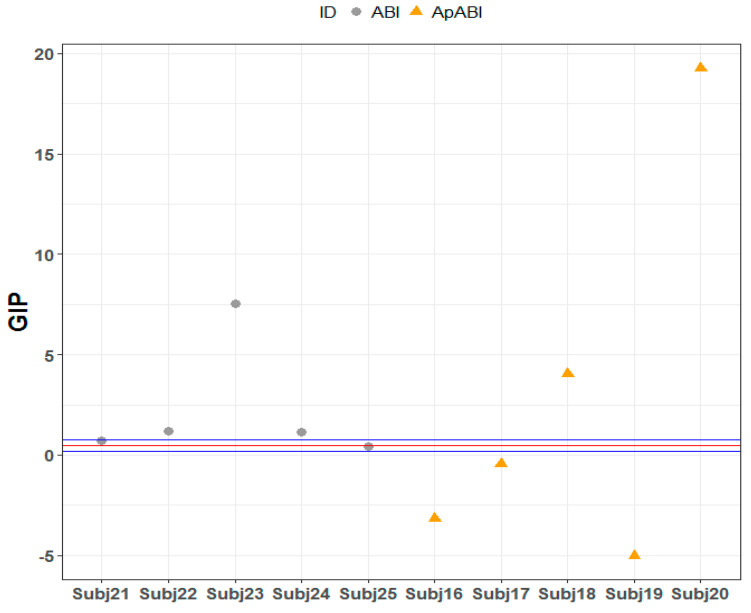
L-FT: GIP mean value of the difference between incongruent and congruent stimuli of each ABI patient, distributed above, below, or within the 95% confidence interval of control group. Red line = GIP mean value of the difference between incongruent and congruent stimuli of the control group; blue line = confidence limits, the two extreme values of the confidence interval which define the range.

**Figure 8 brainsci-13-00298-f008:**
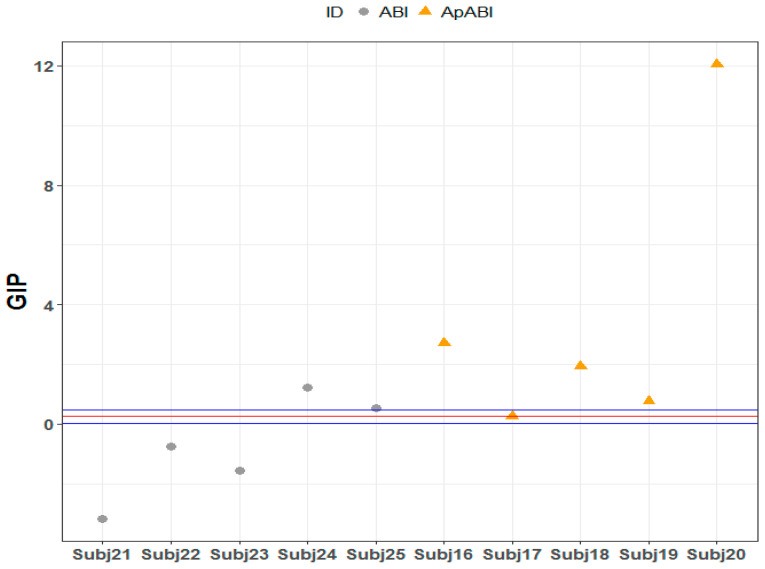
F-FT: GIP mean value of the difference between incongruent and congruent stimuli of each ABI patient, distributed above, below, or within the 95% confidence interval of control group. Red line = GIP mean value of the difference between incongruent and congruent stimuli of the control group; blue line = confidence limits, the two extreme values of the confidence interval which define the range.

**Figure 9 brainsci-13-00298-f009:**
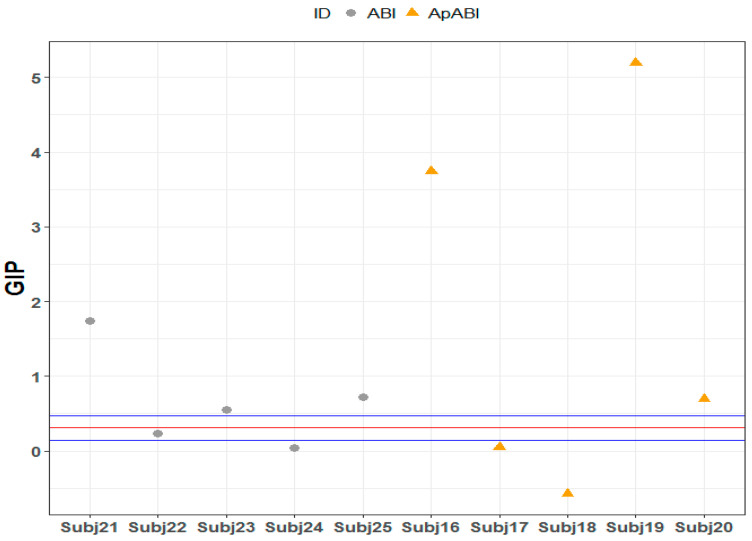
H-FT: GIP mean value of the difference between incongruent and congruent stimuli of each ABI patient, distributed above, below, or within the 95% confidence interval of control group. Red line = GIP mean value of the difference between incongruent and congruent stimuli of the control group; blue line = confidence limits, the two extreme values of the confidence interval which define the range.

**Table 1 brainsci-13-00298-t001:** Clinical and demographic data of participants.

Demographic and Clinical Data	ApABI (n = 5)	ABI (n = 5)	*p*-Value
Age (years)	56.6 ± 12.1	58.6 ± 11.5	0.690
Education (years)	10.4 ± 3.6	16.0 ± 4.4	0.019
Etiology (TBI/non-TBI)	4/1	3/2	0.490
Time since injury (months)	60.4 ± 64.1	34.8 ± 23.7	0.426

Note: ApABI = acquired brain injury patients with diagnosis of apathy; ABI = acquired brain injury patients without diagnosis of apathy; TBI = traumatic brain injury. Statistically significant *p*-values are in bold.

**Table 2 brainsci-13-00298-t002:** Neuropsychological assessment and mood assessment.

Ability/Mood Assessed	Task	ApABI (n = 5)	ABI (n = 5)	*p*-Value
Abstract Reasoning	Raven 36	29.5 ± 3.2	30.8 ± 2.0	0.291
Memory	Digit Span Forward	5.3 ± 0.5	5.1 ± 1.6	0.400
	Digit Span Backward	3.3 ± 1.6	3.7 ± 3.1	0.629
	Prose Memory Test	4.7 ± 2.2	4.9 ± 5.4	0.857
	Corsi Block-Tapping Test (span)	4.2 ± 0.7	4.6 ± 1.5	0.886
Attention	Trail Making Test A	87.5 ± 39.5	44.4 ± 18.6	0.032
	Trail Making Test B	281 ± 92	142 ± 95	0.063
Language	Phonemic Verbal Fluency test	12.2 ± 8.3	25.3 ± 21.0	0.999
	Semantic Verbal Fluency test	10.2 ± 4.2	10.5 ± 3.0	0.857
Executive Functions	Frontal Assessment Battery	15.5 ± 0.5	15.7 ± 1.9	0.670
Depression	Beck Depression Inventory II	19.4 ± 8.9	11.8 ± 7.1	0.151

Note: ApABI = acquired brain injury patients with diagnosis of apathy; ABI = acquired brain injury patients without diagnosis of apathy. Patients’ performances are expressed in mean and standard deviation of corrected scores. Pathological scores are in bold. *p*-Values are obtained from Mann–Whitney U test (or chi-squared test for etiology), in bold if statistically significant.

**Table 3 brainsci-13-00298-t003:** Groups’ characteristics.

Groups	Gender M/F	Age Mean (SD)	Educational Level Mean (SD)
ApABI Patients	3/2	56.6 (12.5)	10.4 (3.36)
ABI Patients without Apathy	3/2	58.6 (11.4)	16 (4.47)
Healthy Subjects	9/6	54.4 (9.57)	13.6 (2.26)

Note: ApABI = acquired brain injury patients with diagnosis of apathy; ABI = acquired brain injury; M = male; F = female; SD = standard deviation.

**Table 4 brainsci-13-00298-t004:** Assessment of apathy in ApABI and ABI patients, and healthy subjects.

Clinical Assessment of Apathy	ApABI Patients	ABI Patients	Healthy Subjects	*p*-Value
Apathy	35.8 ± 0.8 *^§^	59.8 ± 4.3	62.2 ± 3.4	0.002
Cognitive Apathy	16.6 ± 1.8 *^§^	26.4 ± 1.8	27.1 ± 2.4	0.002
Emotional–Affective Apathy	9.6 ± 1.3 *^§^	17.4 ± 1.8	17.3 ± 1.0	0.002
Auto-Activation Apathy	4.6 ± 0.9 *	6.6 ± 1.1	7.3 ± 0.9	0.003

Note: ApABI = acquired brain injury patients with diagnosis of apathy; ABI = acquired brain injury. Data are reported in terms of mean and standard deviation and *p*-values refer to Kruskall–Wallis analysis. Stars indicate a significant difference from healthy subjects, and ^§^ a significant difference from ApABI and ABI patients, all detected with post hoc analysis (U test performed with Bonferroni correction). No differences were detected between ABI patients and healthy subjects. Statistically significant *p*-values are in bold.

**Table 5 brainsci-13-00298-t005:** Correlations between apathy subtypes and performance of subjects.

Type of Task	Congruent Trials			Incongruent Trials		
GIP	Letter FT	Face FT	Hand FT	Letter FT	Face FT	Hand FT
Cognitive Apathy	R = −0.318 *p* = 0.058	R = −0.294 *p* = 0.154	R = −0.252 *p* = 0.224	R = −0.287 *p* = 0.165	R = −0.363 *p* = 0.075	R = −0.245 *p* = 0.070
Emotional–Affective Apathy	**R = −0.479** ***p* = 0.015**	R = −0.343 *p* = 0.094	**R = −0.501** ***p* = 0.011**	**R = −0.407** ***p* = 0.044**	**R = −0.420** ***p* = 0.037**	**R = −0.468** ***p* = 0.018**
Auto-Activation Apathy	R = −0.344 *p* = 0.092	R = −0.300 *p* = 0.145	**R = −0.403** ***p* = 0.046**	R = −0.370 *p* = 0.069	R = −0.301 *p* = 0.144	R = −0.382 *p* = 0.059

Note: Spearman’s correlation coefficient (R) and relevant *p*-values computed between clinical assessment of apathy of all subjects and their task performance (GIP), both in congruent and incongruent trials. Values are reported in bold if statistically significant. The gray cells of the table graphically show the secondary hypothesis of this study: to find correlations between the subtype of apathy and its related task.

**Table 6 brainsci-13-00298-t006:** Patients’ performance in L-FT.

Participants	Statistic	dof	*p*-Value	ES
Subj16	7.640	14	**0.000**	−9.488
Subj17	1.740	14	0.104	−1.945
Subj18	5.627	14	**0.000**	6.660
Subj19	11.781	14	**0.000**	−16.528
Subj20	19.897	14	**0.000**	36.307
Subj21	0.099	14	0.923	−0.109
Subj22	1.327	14	0.206	1.479
Subj23	9.218	14	**0.000**	11.959
Subj24	0.971	14	0.348	1.080
Subj25	0.133	14	0.896	−0.148

Note: Patients’ performance was compared with that of healthy control subjects by using the Crawford–Howell *t*-test for differences between incongruent and congruent stimuli. Subj = subject; dof = degree of freedom; ES = effect size. Statistically significant *p*-values are in bold.

**Table 7 brainsci-13-00298-t007:** Patients’ performance in F-FT.

Participants	Statistic	dof	*p*-Value	ES	Group	p.adj	
Subj21	11.909	14	**0.000**	−16.817	ABI	0.000	***
Subj22	4.984	14	**0.000**	−5.827	ABI	0.001	**
Subj23	7.841	14	**0.000**	−9.805	ABI	0.000	***
Subj24	0.772	14	0.453	0.859	ABI	0.945	
Subj25	1.042	14	0.315	1.160	ABI	0.945	
Subj16	4.679	14	**0.000**	5.439	ApABI	0.002	**
Subj17	0.354	14	0.729	0.393	ApABI	0.945	
Subj18	2.103	14	0.054	2.359	ApABI	0.216	
Subj19	12.147	14	**0.000**	−17.286	ApABI	0.000	***
Subj20	14.444	14	**0.000**	22.154	ApABI	0.000	***

Note: Patients’ performance was compared with that of healthy control subjects, by using the Crawford–Howell *t*-test for differences between incongruent and congruent stimuli. Subj = subject; dof = degree of freedom; ES = effect size. Statistically significant *p*-values are in bold. ** stand for *p* ≤ 0.01; *** stand for *p* ≤ 0.001

**Table 8 brainsci-13-00298-t008:** Patients’ performance in H-FT.

Participants	Statistic	dof	*p*-Value	ES	Group	p.adj	
Subj16	8.954	14	**0.000**	11.548	ApABI	0.000	***
Subj17	0.744	14	0.469	−0.828	ApABI	1.000	
Subj18	3.403	14	**0.004**	−3.874	ApABI	0.030	*
Subj19	10.474	14	**0.000**	14.136	ApABI	0.000	***
Subj20	0.279	14	0.784	−0.310	ApABI	1.000	
Subj21	4.167	14	**0.001**	4.801	ABI	0.008	**
Subj22	0.444	14	0.664	−0.494	ABI	1.000	
Subj23	0.173	14	0.865	0.192	ABI	1.000	
Subj24	1.026	14	0.322	−1.142	ABI	1.000	
Subj25	1.289	14	0.218	1.437	ABI	1.000	

Note: Patients’ performance was compared with that of healthy control subjects by using the Crawford–Howell *t*-test for differences between incongruent and congruent stimuli. Subj = subject; dof = degree of freedom; ES = effect size. Statistically significant *p*-values are in bold. * stands for *p* ≤ 0.05; ** stand for *p* ≤ 0.01; *** stand for *p* ≤ 0.001

## Data Availability

Data are available upon request to the corresponding author.

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
