# Peer review of "The Possible Role of Apathy on Conflict Monitoring: Preliminary Findings of a Behavioral Study on Severe Acquired Brain Injury Patients Using Flanker Tasks"

_brainsci, 2023, doi:10.3390/brainsci13020298_

Round 1

Reviewer 1 Report

Comments and Suggestions for Authors

This study addresses the important issue of clarifying the correlates of apathy in people with acquired brain injury (ABI) by investigating the association between apathy and conflict monitoring abilities. The authors used 2 approaches: group comparison (comparing 2 groups of patients with ABI, 5 with and 5 without apathy, and a group of matched healthy controls) and single-case study, to compare each patient individually with the control group. Patients with apathy had worse performance, driven by a higher number of missing trials, while patients without apathy showed longer reaction times on tasks of conflict monitoring. Associations between performance of different types of tasks and apathy subdomains were also observed, however the small sample size prevents definitive conclusions on the possible specific mechanisms involved.

Here are some suggestions for the paper:

-       + In general, a linguistic check should be carried out on the manuscript to make sure grammar mistakes are amended;

-        + No information is provided about the aetiology of ABI: were all the patients recruited affected by similar brain injuries, e.g. TBI or stroke? I think this is an important aspect that needs clarifying, since also a single-case approach was used, more clinical details should be included in a supplementary file;

-        + Results and tables reported in the “Materials and Methods” section should be moved to the results;

-        + Details on how healthy participants were recruited should be provided;

-        + Line 310: I believe the sentence is referred to “apathy severity”? I think this should be clarified also in the methods (lines 305-307): what scores were correlated with the GIP index (results in Table 4)?

-        + Line 422: which table is Table 3.4?

-        + I think the single-subject analysis is very interesting, given the very restricted sample seize. However, were any of the deficits on conflict monitoring tasks (as reported in Tables 5-7) observed in patients with specific apathy subtypes?

-        + Lines 519-526: results are not so clearly defined at patient-level: also some patients without apathy showed similar conflict monitoring deficits to those observed in patients with apathy and, additionally, not all patients with apathy showed clear deficits;

-        + Line 562-567: the spatial properties (left-right pointing) characterising the stimuli of the hand flanker task represent, indeed, a limitation of the task. This characteristic may have caused some degree of spatial orienting of attention (not present in the other conditions) with a potential confounding effect on performance of patients. Other stimuli could have been selected, e.g. a hand performing the ok sign and hand with a thumb up (both with a similar meaning but clearly different from one another); therefore, this limitation should be stressed in the manuscript;

-        + Another limitation to add to the discussion is the lack of information regarding the type and extent of neurological damage. It is interesting to note that apathetic patients seemed to exhibit greater visuo-spatial attentional/processing speed deficits (i.e., TMT-A completion time) than patients without apathy. This seems to suggest a fronto-parietal(-subcortical) impairment that may contribute to apathy and, potentially, to different apathy subtypes differentially (e.g., greater impact on cognitive apathy).

Reviewer 2 Report

Comments and Suggestions for Authors

The research topic is interesting because apathy is a common disorder after brain injury.

However, the manuscript needs a major of revision.

Abstract:

Remove the italics; give specific information about the sample size and participants (number, mean age) in the control and comparable groups; give brief information about research methods; specify specific results.

Introduction:

Remove the italicized text; remove redundant information of an academic (educational) nature; add information about unresolved problems that caused the choice of the purpose of your research.

Line 32: instead of "traumatic brain injury (TBI)" it is better to write "brain injury" here and further.

Line 33: instead of "Arnould and coll." it is better to write "Arnould et al." here and in the text.

Materials and methods:

I am concerned about the small sample size; it is necessary to explain how the sample size was calculated; add formulas for calculating the minimum sample size in this clinical study.

I recommend moving the specific information about participants and tables to the Results section.

Correct the spelling format, for example: "p-value = 0.024" instead of "p=0.024".

The name of Table 1 needs correction. Transfer all unnecessary details under the table to the Notes.

In Table 2, correct the spelling of the mean (for example, 10,4 (3,36) is non correct).

Results:

Delete italicized words in the text; the names of all tables need correction; notes from the names of tables need to be moved under tables; the names of columns in tables need correction; all columns in tables should have a name.

The presentation of the results in the text is difficult to perceive, it is necessary to improve the style.

In the Discussion, it should be noted how this study differs from the previous ones.

I recommend adding the section Limitation.

Round 2

Reviewer 2 Report

Comments and Suggestions for Authors

The manuscript has been modified and improved by the authors. However, several technical issues were resolved incorrectly or not in full by them.

The names of all tables should be moved above the tables, and notes should be placed under the tables. For example, it is better to leave the name of Table 1 "Clinical and demographic data of participants".

Place a note under the table and write down all abbreviations used in the table. For example, after Table 1: Note: API - ……….; ApABI - ………..; TBI - ………

Above table 2, place only its name "Neuropsychological assessment and mood assessment". Place a note under the table and write down all abbreviations used in the table.

After Table 2: Note: API - ……….; ApABI - ………… Patients performances are expressed in mean and standard deviation of corrected scores. Pathological scores are in bold. P-values are obtained from Mann-Whitney u-test (or chi-squared test for aetiology), in bold if statistically significant.

The names of all tables should be short and clear, place the names above the tables. Place all notes under the tables, as shown in the example of Table 1 and Table 2.

In addition, the name of the first column is missing in Table 5 and Table 6. Add it, please. For example, "Participants".

There are a lot of old references in References. I recommend updating links with a publication age of no more than 5 years (maximum - 10 years).
